# Effects of taurine supplementation on metabolic health and biological aging in healthcare workers: A protocol for a triple-blinded, Bayesian-optimized phase II randomized controlled trial

**Mandy H. M. Chu** [1], **Jacky K. M. Lai** [1], **Anna Lee**[1], **William K. K. Wu**[1], **Ziheng Huang**[1], **Henry M. K. Wong** [1], **Laptin Ho**[1], **Hao Su**[1], **Samantha S. M. Ho**[1], **Xinbo Xu**[1], **Warren Pavey**[2], **David J. R. Morgan** [2,3], **Matthew T. V. Chan** [1], **Kwok M. Ho** [1*]

1 Department of Anaesthesia & Intensive Care, The Chinese University of Hong Kong and Prince of Wales Hospital, Shatin, Hong Kong SAR, China, 2 Institute of Heart and Lung Research Institute, Perth, Western Australia, Australia, 3 Department of Intensive Care Medicine, Fiona Stanley Hospital, Perth, Western Australia, Australia

* kmho@cuhk.edu.hk

## Abstract

### Background

Metabolic-related diseases become increasingly prevalent with age. Recent experimental evidence suggests that taurine (2-aminoethanesulfonic acid) deficiency contributes to these conditions, despite taurine being classified as a conditionally essential amino acid.

### Purpose

This study aims to assess whether a 6-month oral taurine supplementation program improves blood glucose control and other health parameters among healthcare workers.

### Methods

This study is a Bayesian-optimized phase II 1:1 randomized controlled trial. Participants will be randomly assigned to receive oral taurine (3 g/day) or an indistinguishable placebo for 6 months, stratified by (a) diabetes mellitus status and (b) age > 45 years. Assuming *non-informative priors*, posterior probabilities of effectiveness in reducing glycated hemoglobin (HbA1c) will be evaluated after enrollment of 20, 40, and 60 participants, if outcome data are available at those timepoints, to determine whether the trial should be stopped early for futility or superiority, prior to the planned total enrollment of 80 participants (protocol 1.4 on 19th September 2025).

**Data availability statement:** No datasets were generated or analysed during the current study protocol. Study investigators will retain primary rights to the use of the study data. Upon reasonable request to the corresponding author, the study protocol, as well as de-identified raw and summary data, will be made available. In addition, the original, unprocessed data will be published in Mendeley Data following publication of the study results. Data sharing will comply with standard confidentiality requirements; no information that could identify individuals, compromise national security, or interfere with legal processes will be disclosed.

**Funding:** The author(s) received no specific funding for this work.

**Competing interests:** The authors have declared that no competing interests exist.

### Outcomes

The primary outcome is the proportion of participants achieving any reduction in HbA1c at 6 months compared with baseline. Secondary outcomes include changes in plasma lipid levels, blood pressure, PhenoAge, body weight, and skin autofluorescence index.

### Discussion

This phase II trial applies a Bayesian-optimized design to evaluate the potential health benefits of oral taurine supplementation. The findings will inform whether a sufficiently powered phase III randomized controlled trial is warranted to define the role of taurine in promoting metabolic health.

### Trial registration

This trial is registered with the Chinese Clinical Trial registry (https://www.chictr.org.cn/hvshowprojectEN.html?id=277124&v=1.02025-05-21; Identifier: ChiCTR2500102879).

### 1. Introduction

Metabolic-related conditions, including dysglycemia, dyslipidemia, and hypertension, are increasingly prevalent worldwide and represent major contributors to morbidity and healthcare burden. These conditions can develop across the lifespan and are closely linked to lifestyle factors, including diet, physical inactivity, and obesity [1]. Importantly, metabolic dysfunction may precede overt disease and is increasingly recognized as a key determinant of long-term health outcomes.

At the biological level, metabolic dysregulation is associated with multiple underlying mechanisms, including mitochondrial dysfunction, chronic low-grade inflammation, endoplasmic reticulum (ER) stress, and the accumulation of advanced glycation end-products (AGEs), which can contribute to accelerated biological aging [2–5]. These processes contribute not only to the development of chronic diseases but also to reduced physiological reserve. Importantly, growing evidence indicates that these biological changes may be modifiable through targeted interventions.

Lifestyle modification remains the cornerstone of improving metabolic health [6,7]. Regular exercise has consistently been shown to improve glycemic control, cardiovascular health, and overall metabolic function [8–10]. However, emerging evidence suggests that nutritional interventions may play an equally, if not more, important role in modulating metabolic outcomes. For instance, dietary interventions have demonstrated significant effects on glycemic control and cardiometabolic risk factors, sometimes exceeding the benefits observed with exercise alone in certain populations [11]. These findings highlight the potential importance of nutritional strategies, including targeted supplementation [12], in improving metabolic health.

Taurine (2-aminoethanesulfonic acid), traditionally considered a conditionally essential amino acid, is present in high concentrations within cells of energy-demanding organs such as the brain, retina, heart, and skeletal muscle. Intracellular taurine concentrations (5–50 mM) are substantially higher than those in plasma (approximately 100 μM). Taurine levels decline in aging animal and cellular models [13,14], although plasma levels may be maintained in response to hormetic stress [15]. Taurine is considered essential in infants, as deficiency impairs brain development [16]; human milk contains taurine, and infant formulas are supplemented to compensate for limited endogenous synthesis. In adults, taurine is primarily obtained through dietary intake, particularly from seafood and dark meat, with estimated daily consumption rarely exceeding 400 mg. Endogenous synthesis occurs only in limited amounts. Notably, circulating taurine (and N-acetyltaurine) levels can increase following short-term exercise [12], suggesting that taurine may partly mediate the metabolic health benefits of exercise [12,17]. Among these strategies, taurine supplementation has emerged as a promising candidate due to its potential role in metabolic regulation and cellular homeostasis [12].

## 1.1. What we know from prior research

Given the emerging role of nutritional interventions in metabolic health, taurine has attracted increasing research interest. A recent multinational collaborative animal study found that taurine supplementation increased lifespan of middle-aged mice by 10–12% and improved health parameters in older nonhuman primates (equivalent to 45-year-old humans) including lower fat mass, improved bone density, a 19% reduction in fasting blood glucose, and reduced markers of mitochondrial and DNA damage. By improving multiple cellular and organ functions, taurine has been labelled as one of the important amino acids for health [18]. A meta-analysis of human studies showed that taurine supplementation reduced glycated hemoglobin (HbA1c) level by 0.4% [19]. This corresponds to an average reduction in blood glucose of 0.64 mmol/L (or 11.2 mg/dL). Despite this promising result, the validity of this systematic review was limited by the small number of patients included in the analysis (N = 209 from five trials), the short supplementation duration (<16 weeks) in the pooled studies and, most importantly, the largest RCT (N = 63) included also had a high risk of bias in random-sequence generation and allocation concealment [19]. Subsequent to the publication of this review, a RCT (N = 120) involving diabetic patients showed that 8 weeks of taurine supplementation (1g/day) resulted in lower serum insulin levels, Homeostatic Model Assessment of Insulin Resistance (HOMA-IR), and biomarkers of endothelial dysfunction, but not HbA1c [20]. Whether the lack of improvement in HbA1c is due to the short duration of taurine supplementation relative to the long half-life of HbA1c remains uncertain.

We and others have recently reviewed the mechanisms through which taurine may improve health beyond improvements in blood glucose control, including exercise capacity and myocardial function [21,22]. Taurine may exert pleiotropic effects, including improvements in mitochondrial and ER function, pancreatic β–cells survival, bone genesis, and retinal and mental health in different animal models. In addition, taurine may exert its benefits through interactions with the gut microbiome and bile acid conjugation [23], stimulation of colonic glucagon-like-peptide-1 (GLP-1) secretion, and non-competitive inhibition of sodium-glucose transporter-1 (SGLT-1) (with 80% reductions in both $V_{max}$ and Michaelis-Menten constant ($K_m$) for glucose) [24–28]. These potential actions of taurine are important because gut microbiome and bile acids are widely considered as key mediators of human cardiometabolic health [29,30], and SGLT-1 inhibition may reduce small bowel glucose absorption [26] and postprandial hyperglycemia [31]. Furthermore, taurine acts as a reactive carbonyl species (RCS) scavenger through a competitive Schiff-base reaction between RCSs' carbonyl and taurine's amine sites [32–34], reducing glycation of hemoglobin [34] and plasma levels of AGEs [35] as demonstrated in a small RCT recently. Because reducing AGEs accumulation is increasingly being recognized as an important target to improve human long-term cardiometabolic health [5], this potential benefit of taurine should be further evaluated in clinical trials.

## 1.2. Innovation

There is a need to identify innovative strategies to improve health span—not just lifespan—by targeting metabolic health, which may help forestall accelerated or pathological biological aging [36]. Despite the strong scientific foundation

supportive of taurine's health benefits in animal studies, it is important to note that strong experimental data derived from nonhuman animals may not be generalizable to humans. Many nonhuman animals used in experimental studies are not smaller versions of humans because they have vast fundamental physiological differences from humans [37], and are studied under tightly controlled conditions. As such, we must not only continue to conduct high-quality experimental studies, but also validate taurine's promising experimental results in humans.

Taurine is currently synthesized commercially in a pure form (hence also suitable for vegans), considered safe, and inexpensive (<US$1 per gram). Before an adequately powered RCT assessing the effectiveness of taurine in improving long-term human health is conducted, a high-quality phase II trial assessing the medium-term metabolic benefits of taurine supplementation is needed.

Using a Bayesian approach to handle uncertainties is intuitive; and a flexible Bayesian optimized design (BOP2) phase II clinical trial is appealing because at each interim analysis, the futility/superiority decision is made by evaluating a set of posterior probabilities of the events of interest, which is optimized either to maximize power or minimize the number of patients under the null hypothesis (https://www.trialdesign.org/one-page-shell.html#rBOP2) [38,39]. This approach also allows us to explicitly define the type I error rate and the stopping boundary prior to initiation of the trial. This approach offers advantages over post-hoc Bayesian reanalysis of indeterminate results of an RCT [40].

### 1.3. Study aims and hypotheses

We aim to determine whether, compared with placebo, oral taurine supplementation (3 g/day) – a pharmacological dose above the standard dietary intake (<400 mg/day) [41] – over 6 months will: (a) improve glucose metabolism, as reflected by reductions in HbA1c, and (b) improve cardiometabolic parameters, including low-density-lipoprotein (LDL) cholesterol, triglycerides, blood pressure, biological age, and accumulation of AGEs, in adult healthcare workers.

HbA1c was selected as the primary endpoint because it is a clinically meaningful marker of long-term glycemic control, reflecting average blood glucose over 8–12 weeks, and is therefore more appropriate than short-term glycemic measures for evaluating interventions of this duration.

We hypothesize that taurine supplementation will improve glucose metabolism (resulting in reduced HbA1c), and favorably modulate cardiometabolic risk factors, including lipid levels, blood pressure, biological age, and AGE accumulation.

## 2. Methods and materials (Spirit checklist and approved protocols are listed in the Supplementary materials)

### 2.1. Study design and population

The proposed study – abbreviated as **T**aurine **O**r **P**lacebo (**TOP**) Healthcare Worker trial – is a stratified, parallel-arm design, phase II Bayesian-optimized, triple-blinded RCT. Our proposed RCT aims to overcome the existing studies' limitations, and confirm the mechanisms through which taurine could exert its health benefits, as described in our review [21]. After obtaining written informed consent, participants will be randomly allocated to receive either oral taurine or identical-looking capsules daily for a 6-month period in a 1:1 allocation ratio using a computer-generated sequence (with variable block sizes), stratified by chronological age (≤45 vs > 45-years old) and pre-existing diabetic status. Participants will be excluded if they (a) are already taking taurine; (b) have chronic renal disease (eGFR < 30 mL/min) [42]; (c) have insulin-dependent diabetes mellitus; (d) have a bleeding disorder (e.g., platelet count <100 x 10^9/L or require dual anti-platelet therapy [43]); (e) have received active chemo/immunotherapy <30 days prior to enrollment or are likely to receive such treatment during the study period; or (f) are pregnant or planning pregnancy.

Having a healthy workforce is paramount in achieving a sustainable healthcare system, as noted in many countries during the COVID-19 pandemic. This RCT will primarily involve healthcare workers affiliated with the New Territories East Cluster (NTEC) of public hospitals in Hong Kong, with a target of 80 participants over one year. Given their intrinsic motivation to maintain their health, healthcare workers were involved in the design of this trial so that they could adhere to

the study protocol. All healthcare-related personnel who can read and understand either English or the Chinese-translated versions of the study questionnaires – International Physical Activity Questionnaire (IPAQ) and Center for Epidemiologic Studies Depression Scale (CESD-R) [44,45] – will be invited to participate. Eligible participants include doctors, nurses, physiotherapists, pharmacists, dietitians, health administrators, and research-related staff. This study was approved by the NTEC ethics committee on April 2, 2025 (CRE-2024.337-T). The study began enrolling participants in mid-July 2025 and is expected to complete enrollment within 6–9 months, with results anticipated by December 2026.

## 2.2. Blinding and intervention

This study is triple-blinded: observers, statisticians, and participants are all masked to the intervention assignments. This blinding is maintained through the use of taurine and an indistinguishable placebo (high-purity, spray-dried, National Formulary standard, lactose monohydrate). Both were encapsulated by an independent, GCP-certified pharmaceutical company in Western Australia (https://optimaovest.squarespace.com/). Only the pharmacist at this company is aware of the contents of each bottle of study capsules. The REDCap database will be used to generate variable block-size stratified randomization, allocate treatment intervention, and collect data with data quality procedures coded. Both taurine and the indistinguishable placebo capsules are white in color and weigh one gram per capsule. Participants will be advised to take the study capsules in the morning, but participants may take them at other times if necessary. At the end of study, unused study capsules will be counted to assess compliance. Participants will be allowed to eat their normal diet or take their usual medications but will be advised to take the study capsules one hour apart from their medications, even though significant interactions between taurine and commonly used medications have not been reported.

A human pharmacokinetic study showed that after taking 4 grams of taurine (32 mmol) orally, plasma taurine levels peaked at 86 mg/L (0.7 mmol/L) at 1.5 hours [46], exceeding the physiological plasma taurine levels (0.01–0.1 mmol/L) [47] needed to exert its pharmacological benefits [40]. We selected a slightly lower taurine dose in this study because the body weights of most of our study participants, who are mostly Hong Kong Chinese, would be lower than that reported in the pharmacokinetic study (median 79.5 kg) [46].

## 2.3. Enrollment and data collection

After confirming eligibility and obtaining informed consent from prospective participants by the research nurses, baseline data, including fasting blood tests constituting our primary and secondary outcomes, will be collected (Fig 1 and Table 1). Gut microbiome metagenomic analysis is an optional component of the study. The same assessments and fasting blood tests will be repeated at 6 months. A designated research nurse will collect the data from the participants.

## 2.4. Participant incentives

Participants will receive their blood test results, including estimated biological age, and will be informed of their allocation (taurine or placebo) at the end of the study. Transportation costs (taxi fares up to HK$100 [~US$13]) will be reimbursed for participants requiring travel to the study center. No additional financial incentives or compensation will be provided.

## 2.5. Safety monitoring

A two-member independent data safety monitoring committee (DSMC) independent from the sponsor and funder will be formed. Previous human RCTs did not report significant increases in adverse events (AEs) with taurine [19,20,35]. A pre-specified safety threshold will trigger DSMC review. Unblinding to the DSMC will occur if the cumulative incidence of adverse events (AEs) differs significantly between groups, defined as a relative risk >3, with more than 20 cumulative AEs observed in the taurine group. If the AE rate is higher in the taurine group, the trial will be stopped. If the AE rate favors the taurine group, the trial will continue. The reportable AEs include (a) gastrointestinal upset requiring medical attention

| | TRIAL PERIOD (mid-July 2025 to December 2026) | | | |
|---|---|---|---|---|
| | Enrollment (from mid-July 2025) | Post-randomization | | Close-out |
| TIMEPOINT[b] | -$t_i$ to 0 | 0 | 6 months | At 6 months follow-up |
| ENROLLMENT: | | | | |
| Eligibility screen | X | | | |
| Informed consent | X | | | |
| | | | | |
| Randomization | | X | | X |
| INTERVENTION/ COMPARATOR: | | | | |
| [Taurine oral 3grams per day][c] | | X ——————————————→ | | → |
| [Placebo 3 grams per day][d] | | X ——————————————→ | | → |
| ASSESSMENTS: | | X | | X |
| [Age Sex Occupation Chronic medical illnesses Long-term medications Night shift work (never, current within the past 3 months, or ceased for > 3 months) Height (cm) Weight (kg) Body mass index (kg/m2) Systolic blood pressure Diastolic blood pressure] | | X | | X |
| [Hemoglobin A1c, and insulin levels (ELISA) for HOMA-IR (Homeostatic Model Assessment of Insulin Resistance) Low-density-lipoprotein (LDL) Triglyceride Levine PhenoAge, (fasting) glucose]] | | X | | X |
| [(a) Non-invasive skin autofluorescence (by the AGE® reader) (b) Plasma methylglyoxal level (by ELISA) International Physical Activity Questionnaire (IPAQ) Center for Epidemiologic Studies Depression Scale (CESD-R) Brain-derived neurotrophic factor (BDNF) (by ELISA) Shotgun metagenomic sequencing of the fecal sample Plasma taurine and tauroursodeoxycholic acid concentrations (by ELISA)] | | | | |

[a] Recommended content can be displayed using various schematic formats.
[b] List target timepoints and acceptable time windows in this row (e.g., 30 ± 3 days).
[c] Arrow indicates continuous delivery of intervention (e.g., drug)
[d] Example illustrates delivery of comparator at discrete timepoints (e.g., psychotherapy)

Citation: Chan A-W, Boutron I, Hopewell S, Moher D, Schulz KF, et al. SPIRIT 2025 statement: updated guideline for protocols of randomised trials. BMJ 2025;389:e081477. https://dx.doi.org/10.1136/bmj-2024-081477

**Fig 1. Participant timeline: Schedule of enrollment, interventions, and assessments of the TOP Healthcare worker Trial.**

**Table 1. Data collection at baseline and 6-month final follow-up.**

| Baseline characteristics | Measure |
|---|---|
| Demographic and baseline health factors | Age |
| | Sex |
| | Occupation |
| | Chronic medical illnesses |
| | Long-term medications |
| | Night shift work (never, current within the past 3 months, or ceased for > 3 months) |
| Body composition | Height (cm) |
| | Weight (kg) |
| | Body mass index (kg/m$^2$) |
| **Secondary outcomes:** | |
| Blood pressure | Systolic blood pressure |
| | Diastolic blood pressure |
| Glucose metabolism | Hemoglobin A1c, and insulin levels (ELISA) for HOMA-IR (Homeostatic Model Assessment of Insulin Resistance) |
| Lipid metabolism | Low-density-lipoprotein (LDL) |
| | Triglyceride |
| Biological or Phenotypical age | Levine PhenoAge (using lymphocyte percentage, mean cell volume, red cell width distribution, total white cell count, albumin, (fasting) glucose, C-reactive protein, alkaline phosphatase, creatinine, and chronological age) [36] |
| **Exploratory outcomes:** | |
| Advanced glycation end-products (AGEs) | Non-invasive skin autofluorescence (by the AGE° reader); plasma methylglyoxal level (by ELISA) |
| Physical activity | International Physical Activity Questionnaire (IPAQ) |
| Mental health | Center for Epidemiologic Studies Depression Scale (CESD-R) Brain-derived neurotrophic factor (BDNF) (by ELISA) |
| Gut microbiome | Shotgun metagenomic sequencing of the fecal sample |
| Taurine intake | Plasma taurine and tauroursodeoxycholic acid concentrations (by ELISA) |

or participant unwilling to resume the study capsules, (b) allergic skin rashes, (c) documented spontaneous hypoglycemia with blood glucose <4 mmol/L, (d) unexpected bleeding from any site after commencement of study capsules, and (e) any new symptoms the participants believe are related to the study medications resulting in cessation of study capsules.

### 2.6. Data sharing plan

Study investigators will retain primary rights to the use of the study data. Upon reasonable request to the corresponding author, the study protocol, as well as de-identified raw and summary data, will be made available. In addition, the original, unprocessed data will be published in Mendeley Data following publication of the study results. Data sharing will comply with standard confidentiality requirements; no information that could identify individuals, compromise national security, or interfere with legal processes will be disclosed.

## 2.7. Primary and secondary outcomes, sample size calculation, and data analysis

Our sample size calculation is based on the **primary outcome**: the proportion of participants achieving any reduction in HbA1c at 6 months compared to baseline. Taurine was previously shown to reduce HbA1c by 0.41% (95%CI 0.09–0.74; p = 0.001) [19]. In this BOP2 trial, stopping criteria for superiority or futility are based on *non-informative priors.* We assume that HbA1c will be reduced after 6 months of intervention in 40% of participants in the experimental group and 10% in the control group (reflecting a potential Hawthorne effect) [48]. The margin of meaningful difference between the two groups is set at 10%. Assuming one-sided type I error of 10% is acceptable, this trial will have 80% statistical power with a total sample size of 80. Assuming no informative prior, posterior probabilities of effectiveness after enrolling 20, 40, and 60 participants will be used to determine whether the trial should be stopped, due to futility or superiority, prior to the completion of enrollment (80 participants). The margins for futility or superiority in the three interim analyses after enrolling 20, 40 and 60 patients are described in the algorithm in the supplementary materials (Appendix I in S1 Appendix). The statistician analyzing the data, based on intention-to-treat approach, will be blinded to the treatment allocation during the interim analyses, unless the trial results meet the margins for futility or superiority.

**Secondary and exploratory outcomes** will assess between-group differences at 6 months across measures of general, gut, mental, and cardiometabolic health (Table 1) – will be assessed using the Mann-Whitney U test and analysis of covariance (ANCOVA), adjusting for baseline values as appropriate. The secondary outcomes include blood pressure, body weight, biological age (measured by the PhenoAge), plasma lipid parameters, HOMA-IR (Homeostatic Model Assessment of Insulin Resistance), advanced glycation end-products (assessed by the non-invasive skin autofluorescence (SAF) and plasma methylglyoxal levels), gut microbiome alpha- and beta-diversity and species-level sub-communities (by shotgun metagenomic analyses), physical activity (measured by the International Physical Activity Questionnaire), and mental health symptoms (measured by the Center for Epidemiologic Studies Depression Scale). These outcomes are evaluated because cardiometabolic health is closely interrelated with biological aging, mental health, circadian disruption (e.g., shift work), and gut microbiome composition [36,49–58].

A subgroup analysis by excluding participants allocated to the taurine group who have 6-month plasma taurine levels less than the mean plasma taurine level of the control group, suggestive of non-compliance to the study intervention, will be conducted. A one-sided p-value <0.10 without Bonferroni adjustment will be taken as significant for the primary outcome in this phase II trial. Continuous outcome measures, described in Table 1, will be analyzed by analysis of covariance (ANCOVA) (while adjusting for baseline data) [59], and a p-value <0.05 without Bonferroni adjustment will be considered significant for these secondary outcome measures. Furthermore, the associations among these continuous outcome measures will be explored to advance our understanding of the mechanisms underlying accelerated biological aging [60].

## 3. Discussion

The aging population represents a significant global challenge, including Hong Kong. As observed during the COVID-19 pandemic in many countries, maintaining a healthy population and a resilient healthcare workforce is essential for a sustainable healthcare system. Beyond aging, a growing number of individuals are living with metabolic syndrome, frailty, or multiple comorbidities, which negatively impact their quality of life and place increasing demands on healthcare services [1]. Many chronic diseases, including metabolic syndrome, are associated with accelerated biological aging [36]. From a public health perspective, this is particularly concerning, as accelerated aging is more prevalent among the minoritized groups due to lifelong exposure to adverse social determinants of health [61]. Inexpensive and accessible interventions such as taurine supplementation that improve metabolic health may therefore have a substantial public health impact.

Our proposed TOP healthcare worker trial will utilize a Bayesian-optimized phase II 1:1 randomized controlled trial design to assess the promising health benefits of taurine supplementation. Taurine has been assessed in some clinical trials and important adverse effects compared to placebo have not been reported [20,62]. We recognize the importance of allocation concealment and blinding to avoid selection bias, and the potential to have a Hawthorne (beneficial) effect in the

placebo group [63]. By conducting an adequately powered phase II trial with *a priori* defined futility and superiority stopping margins, we will be able to generate robust clinical data before we embark on conducting an adequately powered phase III trial to define the role of taurine in promoting healthy aging.

In order to inform the design of future phase III trials, such as what health endpoints should be used, assessment of clinically relevant secondary outcomes is essential. In recent years, accumulation of AGEs [49,64–67], accelerated biological aging [36,68–71], and gut dysbiosis [72] — all potentially modifiable by taurine supplementation — have been reported to contribute to the development of many chronic illnesses, many of which are closely linked to metabolic dysfunction, including depression, frailty, recurrent falls, osteoporosis, complications after major surgery, mortality after critical illness, and neurodegenerative diseases. For example, a recent study showed that accelerated biological aging was common in patients with renal failure, and was quantitatively associated with the accumulation of AGEs (assessed by the non-invasive skin autofluorescence: SAF age = [SAF − 0.83]/0.024; correlation between SAF age and PhenoAge = 0.35, p = 0.02) [73]. This result suggests that 'advanced SAF age' may be used as a non-invasive surrogate marker of accelerated biological aging. In sum, quantifying the relationships among clinically relevant secondary outcomes in this trial will help inform the design of future phase III RCTs evaluating the role of taurine in promoting metabolic health in aging populations.

To our knowledge, our proposed study will be the first RCT that investigates the potential effects of taurine on multiple health domains in humans. Upon completion of enrollment and data analysis, we will disseminate the findings through publications in peer-reviewed clinical journals and presentations in key medical conferences. If the results are promising, a phase III RCT including follow-up beyond 6 months to confirm the long-term health benefits of taurine supplementation for the aging society is warranted. While avoiding death is not possible, improving metabolic health to reduce premature biological decline remains an important goal.

## Supporting information

**S1 Appendix. Appendix I.**
(XLSX)

**S1 File. CREC application TOP Healthcare workers trial protocol.1.4.**
(PDF)

**S1 Checklist. TOP Healthcare worker trial SPIRIT 2025 editable checklist.**
(DOCX)

## Author contributions

**Conceptualization:** Mandy H. M. Chu, Jacky K. M. Lai, Anna Lee, William K. K. Wu, Ziheng Huang, Henry M. K. Wong, Laptin Ho, Hao Su, Samantha S. M. Ho, Xinbo Xu, Warren Pavey, David J. R. Morgan, Matthew T. V. Chan, Kwok M. Ho.

**Data curation:** Mandy H. M. Chu.

**Funding acquisition:** Kwok M. Ho.

**Investigation:** Mandy H. M. Chu, Jacky K. M. Lai, Ziheng Huang, Henry M. K. Wong, Hao Su, Warren Pavey, Matthew T. V. Chan, Kwok M. Ho.

**Methodology:** Mandy H. M. Chu, Anna Lee, William K. K. Wu, Ziheng Huang, Henry M. K. Wong, Laptin Ho, Hao Su, Samantha S. M. Ho, Xinbo Xu, David J. R. Morgan, Matthew T. V. Chan, Kwok M. Ho.

**Project administration:** Mandy H. M. Chu, Kwok M. Ho.

**Writing – original draft:** Mandy H. M. Chu, Jacky K. M. Lai, Anna Lee, William K. K. Wu, Ziheng Huang, Henry M. K. Wong, Laptin Ho, Samantha S. M. Ho, Xinbo Xu, Warren Pavey, David J. R. Morgan, Matthew T. V. Chan, Kwok M. Ho.

**Writing – review & editing:** Mandy H. M. Chu, Jacky K. M. Lai, Anna Lee, William K. K. Wu, Ziheng Huang, Henry M. K. Wong, Laptin Ho, Hao Su, Samantha S. M. Ho, Xinbo Xu, Warren Pavey, David J. R. Morgan, Matthew T. V. Chan, Kwok M. Ho.

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
