## [Decision Letter · Decision Letter 0]

7 May 2026

PONE-D-25-52928Effects of Taurine Supplementation on Metabolic Health and Biological Aging in Healthcare Workers: A Protocol for a Triple-blinded, Bayesian-Optimized Phase II Randomized Controlled TrialPLOS One

Dear Dr. Ho,

Thank you for submitting your manuscript to PLOS ONE. After careful consideration, we feel that it has merit but does not fully meet PLOS ONE’s publication criteria as it currently stands. Therefore, we invite you to submit a revised version of the manuscript that addresses the points raised during the review process.

We look forward to receiving your revised manuscript.

Kind regards,

Diego García-Ayuso, PhD

Academic Editor

PLOS One

**Journal Requirements:**

1. When submitting your revision, we need you to address these additional requirements. Please ensure that your manuscript meets PLOS ONE's style requirements, including those for file naming. The PLOS ONE style templates can be found at https://journals.plos.org/plosone/s/file?id=wjVg/PLOSOne_formatting_sample_main_body.pdf and https://journals.plos.org/plosone/s/file?id=ba62/PLOSOne_formatting_sample_title_authors_affiliations.pdf 2. Please include captions for your Supporting Information files at the end of your manuscript, and update any in-text citations to match accordingly. Please see our Supporting Information guidelines for more information: http://journals.plos.org/plosone/s/supporting-information. 3. If the reviewer comments include a recommendation to cite specific previously published works, please review and evaluate these publications to determine whether they are relevant and should be cited. There is no requirement to cite these works unless the editor has indicated otherwise.

**Additional Editor Comments:**

1. Please revise the Data Availability statement to ensure that it complies with PLOS ONE policy. The current wording indicating that data will be available upon reasonable request should be clarified, including where de-identified participant-level data, data dictionary, statistical code and relevant materials will be deposited or, if restrictions apply, the exact nature of those restrictions.

2. Please ensure that statements are moderated that may imply that the trial will directly assess ageing-related diseases or establish taurine as an intervention for healthy ageing. The study appears to have been primarily designed to evaluate metabolic health markers, biological/phenotypic age indicators and related exploratory outcomes in healthcare workers.

3. Please clarify in the manuscript that, as a study protocol, no trial results are expected at this stage. The abstract "Results" section should be restricted to planned outcomes or anticipated reporting, ensuring that it does not convey the impression that results are already available.

4. Please check the manuscript and supplementary materials for minor issues with wording and formatting. Please ensure that 'Enrolment' is used consistently in Figure 1 and that the apparent stray 'F' in Table 1 under the AGEs section is corrected.

5. It is essential to clearly distinguish all secondary and exploratory outcomes, with particular attention to the microbiome, BDNF, mental health, IPAQ/CESD-R and PhenoAge-related analyses.

Reviewers' comments:

Reviewer's Responses to Questions

**Comments to the Author**

1. Does the manuscript provide a valid rationale for the proposed study, with clearly identified and justified research questions?

Reviewer #1: Yes

Reviewer #2: No

2. Is the protocol technically sound and planned in a manner that will lead to a meaningful outcome and allow testing the stated hypotheses?

Reviewer #1: Yes

Reviewer #2: Yes

3. Is the methodology feasible and described in sufficient detail to allow the work to be replicable?

Reviewer #1: Yes

Reviewer #2: Yes

4. Have the authors described where all data underlying the findings will be made available when the study is complete?

Reviewer #1: No

Reviewer #2: No

5. Is the manuscript presented in an intelligible fashion and written in standard English?

Reviewer #1: Yes

Reviewer #2: No

6. Review Comments to the Author

You may also provide optional suggestions and comments to authors that they might find helpful in planning their study.

**Reviewer #1:** Lovely description of a study, with a clear indication of the statistical Bayesian design used.

Consider Enrolment in Figure 1 rather than Enrollment

**Reviewer #2:** The manuscript entitled “Effects of Taurine Supplementation on Metabolic Health and Biological Aging in Healthcare Workers: A Protocol for a Triple-blinded, Bayesian-Optimized Phase II Randomized Controlled Trial” written by Mandy Hiu Man Chu and colleagues PROPOSES a clinical trial investigating the effects of taurine supplementation on glucose metabolisms in healthcare workers.

Therapeutic avenues offered by taurine are interesting and promising hot scientific topics. However, the manuscript exhibits two main flaws:

1/ This study aims at investigating the effects of taurine supplementation on ageing-related diseases. But ageing is not systematically and strictly associated with a metabolic syndrome. Here, the main outcomes are HbA1c concentrations

plasma lipid levels. But these parameters are more relevant for metabolic syndrome than ageing-related diseases. Furthermore, metabolic syndrome is mainly caused by “the lifestyle” (diet, exercise) than by ageing.

2/ There is no chapter which aims at presenting the results ? This is a scientific study which is a scientific project. Is a Scientific project able to be published in Plos one ? Although authors aim at publishing a “protocol”, is Plos One the relevant journal, and what is original in this protocol project?

Consequently, this manuscript cannot be accepted in this form and could be resubmitted when results are available.

7. PLOS authors have the option to publish the peer review history of their article (what does this mean?). If published, this will include your full peer review and any attached files.

Reviewer #1: No

Reviewer #2: No

---

## [Author Response · Author response to Decision Letter 1]

7 May 2026

May 8, 2026

Diego García-Ayuso, PhD

Academic Editor

PLOS One

Dear Dr García-Ayuso,

Thank you for your decision letter dated May 8, 2026, and for the reviewers’ constructive comments. We have revised the manuscript accordingly. Below, we provide a point-by-point response outlining the changes made.

Additional Editor Comments:

1. Please revise the Data Availability statement to ensure that it complies with PLOS ONE policy. The current wording indicating that data will be available upon reasonable request should be clarified, including where de-identified participant-level data, data dictionary, statistical code and relevant materials will be deposited or, if restrictions apply, the exact nature of those restrictions.

Response: We have revised our data sharing plan as follows: “Study investigators will retain primary rights to the use of the study data. Upon reasonable request to the corresponding author, the study protocol, as well as de-identified raw and summary data, will be made available. In addition, the original, unprocessed data will be published in Mendeley Data following publication of the study results. Data sharing will comply with standard confidentiality requirements; no information that could identify individuals, compromise national security, or interfere with legal processes will be disclosed.”

2. Please ensure that statements are moderated that may imply that the trial will directly assess ageing-related diseases or establish taurine as an intervention for healthy ageing. The study appears to have been primarily designed to evaluate metabolic health markers, biological/phenotypic age indicators and related exploratory outcomes in healthcare workers.

Response: This is an important suggestion. We have revised our abstract as well as different parts of the manuscript, in particular the background, to focus our topic primarily around metabolic health instead of healthy aging in general.

3. Please clarify in the manuscript that, as a study protocol, no trial results are expected at this stage. The abstract "Results" section should be restricted to planned outcomes or anticipated reporting, ensuring that it does not convey the impression that results are already available.

Response: We have changed the word “Results” to “Outcomes” to avoid misleading the readers that results are already available.

4. Please check the manuscript and supplementary materials for minor issues with wording and formatting. Please ensure that 'Enrolment' is used consistently in Figure 1 and that the apparent stray 'F' in Table 1 under the AGEs section is corrected.

Response: We have corrected the error in Table 1:

Advanced glycation end-products (AGEs): Non-invasive skin autofluorescence (by the AGE® reader); plasma methylglyoxal level (by ELISA).

We have not changed the enrollment in Figure 1 to enrolment because we have used American English throughout the manuscript (versus enrolment in British English).

5. It is essential to clearly distinguish all secondary and exploratory outcomes, with particular attention to the microbiome, BDNF, mental health, IPAQ/CESD-R and PhenoAge-related analyses.

Response: We have revised Table 1 to separate secondary and exploratory outcomes in the revised manuscript.

Reviewer's Responses to Questions

Comments to the Author

1. Does the manuscript provide a valid rationale for the proposed study, with clearly identified and justified research questions?

Reviewer #1: Yes

Reviewer #2: No

Response: We have revised the Background section to strengthen the rationale for testing the potential metabolic health benefits of taurine.

2. Is the protocol technically sound and planned in a manner that will lead to a meaningful outcome and allow testing the stated hypotheses?

Reviewer #1: Yes

Reviewer #2: Yes

Response: We understand this is a phase II trial and have separated secondary and exploratory outcomes to suggest that secondary and exploratory outcomes are hypothesis-generating instead of being definitive.

3. Is the methodology feasible and described in sufficient detail to allow the work to be replicable?

Reviewer #1: Yes

Reviewer #2: Yes

Response: We have completed the enrollment of the target 80 participants with only one participant had adverse gastrointestinal side effects requiring cessation of the ongoing intake of the study capsules. The trial intervention remains blinded currently and we are not sure whether this participant was taking placebo or taurine. Because of the fast enrollment rate, we did not have the change to execute interim analysis so far.

4. Have the authors described where all data underlying the findings will be made available when the study is complete?

Reviewer #1: No

Reviewer #2: No

Response: We have revised our data sharing plan as follows: “Study investigators will retain primary rights to the use of the study data. Upon reasonable request to the corresponding author, the study protocol, as well as de-identified raw and summary data, will be made available. In addition, the original, unprocessed data will be published in Mendeley Data following publication of the study results. Data sharing will comply with standard confidentiality requirements; no information that could identify individuals, compromise national security, or interfere with legal processes will be disclosed.”

5. Is the manuscript presented in an intelligible fashion and written in standard English?

Reviewer #1: Yes

Reviewer #2: No

Response: We have revised our manuscript for stylistic and grammatical improvement and we believe this has improved the clarity of the manuscript.

6. Review Comments to the Author

You may also provide optional suggestions and comments to authors that they might find helpful in planning their study.

Reviewer #1: Lovely description of a study, with a clear indication of the statistical Bayesian design used.

Response: Thank you for our time to review our work.

Consider Enrolment in Figure 1 rather than Enrollment

Response: We have not changed the enrollment in Figure 1 to enrolment because we have used American English throughout the manuscript (versus enrolment in British English).

Reviewer #2: The manuscript entitled “Effects of Taurine Supplementation on Metabolic Health and Biological Aging in Healthcare Workers: A Protocol for a Triple-blinded, Bayesian-Optimized Phase II Randomized Controlled Trial” written by Mandy Hiu Man Chu and colleagues PROPOSES a clinical trial investigating the effects of taurine supplementation on glucose metabolisms in healthcare workers.

Therapeutic avenues offered by taurine are interesting and promising hot scientific topics. However, the manuscript exhibits two main flaws:

1/ This study aims at investigating the effects of taurine supplementation on ageing-related diseases. But ageing is not systematically and strictly associated with a metabolic syndrome. Here, the main outcomes are HbA1c concentrations, plasma lipid levels. But these parameters are more relevant for metabolic syndrome than ageing-related diseases. Furthermore, metabolic syndrome is mainly caused by “the lifestyle” (diet, exercise) than by ageing.

Response: This is an important suggestion. We have extensively revised our abstract as well as different parts of the manuscript, in particular the background, to focus our topic primarily around metabolic health instead of healthy aging in general. That said, growing evidence suggests that many lifestyle factors are related to health aging (or a back of it, resulting in accelerated biological aging),

2/ There is no chapter which aims at presenting the results? This is a scientific study which is a scientific project. Is a Scientific project able to be published in Plos one? Although authors aim at publishing a “protocol”, is Plos One the relevant journal, and what is original in this protocol project? Consequently, this manuscript cannot be accepted in this form and could be resubmitted when results are available.

Response: We believe PLOS One specifically welcomes publication of protocols and it is widely recommended to have the study protocol published before analysis of the results for transparency reasons. We expect completion of 6-month follow up of the last enrolled participant by September 2026 and will have the full results by late 2026 or early 2027. It is, therefore, important to publish the trial protocol before September 2026 when we will analyze the results.

7. PLOS authors have the option to publish the peer review history of their article (what does this mean?). If published, this will include your full peer review and any attached files.

Do you want your identity to be public for this peer review? For information about this choice, including consent withdrawal, please see our Privacy Policy.

Reviewer #1: No

Reviewer #2: No

Thank you very much again for all the constructive comments and suggestions which have substantially improved our manuscript.

Yours sincerely,

KM

Professor Kwok M. Ho on behalf of all authors

---

## [Editor Report · Decision Letter 1]

13 May 2026

Effects of Taurine Supplementation on Metabolic Health and Biological Aging in Healthcare Workers: A Protocol for a Triple-blinded, Bayesian-optimized Phase II Randomized Controlled Trial

PONE-D-25-52928R1

Dear Dr. Ho,

We’re pleased to inform you that your manuscript has been judged scientifically suitable for publication and will be formally accepted for publication once it meets all outstanding technical requirements.

Kind regards,

Diego García-Ayuso, PhD

Academic Editor

PLOS One

Additional Editor Comments (optional):

Please consider slightly moderating a few narrative expressions related to longevity and healthy aging (e.g., “longevity amino acids”, “flattening the biological age curve”, “cheating death”), in keeping with the neutral tone generally preferred for clinical trial protocols.
---

## [Editor Report · Acceptance letter]

PONE-D-25-52928R1

PLOS One

Dear Dr. Ho,

I'm pleased to inform you that your manuscript has been deemed suitable for publication in PLOS One. Congratulations! Your manuscript is now being handed over to our production team.

Kind regards,

on behalf of

Dr. Diego García-Ayuso

Academic Editor

PLOS One